# Gene Electrotransfer Efficiency in 2D and 3D Cancer Cell Models Using Different Electroporation Protocols: A Comparative Study

**DOI:** 10.3390/pharmaceutics15031004

**Published:** 2023-03-21

**Authors:** Alexia de Caro, Elisabeth Bellard, Jelena Kolosnjaj-Tabi, Muriel Golzio, Marie-Pierre Rols

**Affiliations:** Institut de Pharmacologie et de Biologie Structurale du CNRS UMR 5089, 205, Route de Narbonne, 31077 Toulouse CEDEX, France

**Keywords:** electropermeabilization, gene electrotransfer, cells suspensions, spheroids, microscopy, clearing

## Abstract

Electroporation, a method relying on a pulsed electric field to induce transient cell membrane permeabilization, can be used as a non-viral method to transfer genes in vitro and in vivo. Such transfer holds great promise for cancer treatment, as it can induce or replace missing or non-functioning genes. Yet, while efficient in vitro, gene-electrotherapy remains challenging in tumors. To assess the differences of gene electrotransfer in respect to applied pulses in multi-dimensional (2D, 3D) cellular organizations, we herein compared pulsed electric field protocols applicable to electrochemotherapy and gene electrotherapy and different “High Voltage–Low Voltage” pulses. Our results show that all protocols can result in efficient permeabilization of 2D- and 3D-grown cells. However, their efficiency for gene delivery varies. The gene-electrotherapy protocol is the most efficient in cell suspensions, with a transfection rate of about 50%. Conversely, despite homogenous permeabilization of the entire 3D structure, none of the tested protocols allowed gene delivery beyond the rims of multicellular spheroids. Taken together, our findings highlight the importance of electric field intensity and the occurrence of cell permeabilization, and underline the significance of pulses’ duration, impacting plasmids’ electrophoretic drag. The latter is sterically hindered in 3D structures and prevents the delivery of genes into spheroids’ core.

## 1. Introduction

The principle of gene electrotransfer relies on the application of electric field pulses used to permeabilize the cell membrane in the presence of nucleic acids, for example, plasmid DNA, which can subsequently penetrate the cell and modify the gene expression. The first gene electrotransfer assay was conducted in 1972 by Neumann and colleagues on mouse lyoma cells, using a high field strength (8 kV/cm) and low pulse time (5 µs) [1]. This approach was subsequently used in vivo on healthy tissues, thanks to works done to transfect DNA in murine skin cells [2], followed by the transfer of antiviral vectors in rat kidneys [3] and plasmid DNA transfer into muscle fibers for the treatment of muscular dysfunctions [4]. In addition, in 1998, Rols and colleagues pioneered the electrotransfer of genes and proteins into cancer tissue, namely, the murine melanoma [5]. These in vivo tests have incited a considerable number of in vitro tests to unveil the mechanisms of gene electrotransfer and to optimize the electrical parameters and electroporation conditions. To name but a few: the effect of serum was studied [6], the effect of cell synchronization was investigated [7], and the effect of polarity and pulse orientation was determined [8]. The mechanism of DNA internalization after electroporation can be described as a succession of different steps, which are under the control of pulses’ parameters [9]: (i) migration of DNA to the membrane by electrophoretic forces, (ii) formation of DNA–membrane complexes [10] that occur during electric pulses and then (iii) cellular internalization [9,11], which can be mediated by endocytosis [12], and (iv) migration of DNA into the nucleus via cytoskeleton [12] and gene expression [6].

Electrotransfer can also be used to induce gene extinction for gene therapy. This electroporation-mediated therapeutic strategy was studied to target a specific glomerulus gene involved in kidney disease [13], but also in the oncology field, where the role of siRNA intracellular traffic was studied in murine melanoma cell lines [14,15]. In addition, Golzio and colleagues studied siRNA delivery in solid murine tumors in 2007 [16] and its potential use in clinics [17].

Thus, these studies led to preclinical [18] and clinical trials. A Phase I human clinical trial involving the administration of siRNA to patients with solid tumors using a targeted nanoparticle delivery system was carried out in 2010 [19]. Moreover, a Phase I clinical trial was conducted using plasmids encoding interleukin IL-12 in patients with electroporated metastatic melanoma. A complete regression was observed, although pain was noted as the main side effect [20]. Compared to clinical electrochemotherapy, where standardized electrical parameters are used [21,22], the electrical parameters for gene transfer are not defined, and there are no standard operating procedures applicable in clinics. However, it has been proven for decades that microsecond pulses used for electrochemotherapy are less effective than millisecond pulses for gene transfer, either in the muscles [4], tumors [5] or liver [23]. In addition, low field values are required to avoid cell death [4,24]. The electrical parameters for gene transfer are, therefore, continuously optimized, and some studies have proven their efficiency, in particular with pulses in the “High Voltage–Low Voltage” (HV-LV) range. Studies reported that short pulses of high intensity, also referred to as “high voltage (HV)”, allow cell permeabilization, and longer pulses of lower intensity, also known as “lower voltage (LV)”, allow DNA migration by electrophoretic drag and, therefore, increase gene expression on CHO cells in vitro [25,26,27] in rat skin [28] and muscle [29,30,31]. In 2010, it was shown that HV-LV pulses allow a higher percentage of transfected cells with a lower concentration of plasmid, in contrast to assays conducted under HV or LV-HV regimen [26]. In addition, it was suggested that bipolar HV-LV pulses could further increase gene expression in vitro compared to unipolar pulses. Finally, more recently, a study of gene electrotransfer was conducted on a 3D collagen gel, suggesting that lower transfection rates were obtained, in contrast to adherent cells [32].

Although a considerable amount of in vitro and in vivo studies was published, only a few studies on gene electrotransfer were conducted in a multicellular spheroid model. Indeed, the spheroid represents a complex multicellular environment that might respond in a heterogenous way to electroporation, in comparison to isolated suspended cells. This was neatly highlighted by Canatella and colleagues, who empirically and theoretically showed that the heterogeneous response of multicellular spheroids to electroporation and calcein uptake might be due to: (1) the difference in cells state (namely, the cells in the core of the spheroid are smaller), (2) the local solute concentration (e.g., as cells in spheroids are tightly packed, the extracellular solute reservoir is limited and smaller than the one in cells in suspension, and the solute has a diffusional lag time), and (3) the local electric field (as neighboring cells may reduce the local electric field), resulting in reduced uptake of calcein in the spheroids’ core following electroporation [33].

A few studies investigated gene electrotransfer in the spheroid model, which closely resembles living tissues. Results obtained in 2009 show that even if cells are effectively permeabilized by electric fields under the GET protocol, when including the cells within spheroids’ core, gene expression remains limited to the outer layers of the spheroid [34]. Similarly to Canatella et al., Wasungu et al. highlighted the difference in the plasmid transfection efficiency in isolated cells suspensions and 3D-grown cells. While suspended cells had a transfection rate of 24%, only 1% of cells were transfected in spheroids, with a field value of 500 V/cm [24]. As the plasmid size might contribute to poor penetration into the spheroids’ core, in 2021, an attempt to transfect spheroids with smaller molecules, namely, the small interfering RNAs (siRNAs), was conducted. Nevertheless, the conclusions were substantially identical; the internalization of siRNA was not homogeneous in the spheroids, which was in complete agreement with in vivo studies [35].

In view of previous investigations, the purpose of this study was to study potential differences in the efficiency and rate of plasmid transfection following the application of different electroporation protocols. Precisely, we compared the standard electrochemotherapy (ECT) protocol; parameters used for electro-genotherapy (GET), which are currently used in laboratory research; and the “High Voltage–Low Voltage” pulses combinations, which could be of particular interest for their potential to permeabilize cells (via their high-voltage component) and improve the penetration of plasmids (thanks to the improved electrophoretic drag, allowed by the low-voltage component). The three protocols were tested on human colon cancer cells in suspension and grown in multicellular spheroids, i.e., in 2D and 3D models. In addition to classical methods of detection of permeabilization and transfection (standard live-cell microscopy, flow cytometry), we used the emerging approach of tissue clearing, which enables deeper 3D imaging of the spheroid across its volume. This novel tissue processing technique could allow arbitrating upon the lack of fluorescence visualization within the spheroids’ core following electroporation, which might be due to (1) permeabilization absence and the concomitant absence of fluorescence dye penetration, or (2) the impossibility of fluorescence detection due to the opacity of the electroporated tissue, located in the spheroids’ core.

## 2. Materials and Methods

### 2.1. Cell Culture and Spheroids Formation

HCT-116- (ATCC^®^ CCL-247™) and HCT-116-expressing, green, fluorescent, protein (GFP) [36] colon cancer cell lines were cultured in a controlled humidified atmosphere (5% CO_2_, 37 °C). Cells proliferated in a Dulbecco’s Modified Eagle Medium (DMEM) cell culture medium with 4.5 g/L of D-glucose, L-Glutamine containing pyruvate (Gibco-Invitrogen, Cergy-Pontoise, France) supplemented with 10% fetal calf serum (Sigma Aldrich, St. Louis, MO, USA) and 1% penicillin/streptomycin (Sigma Aldrich, USA). Spheroids were generated in Costar^®^ Corning^®^ Ultra Low Attachment 96-well plates. Five hundred cells were seeded in 200 µL of medium per well in a humidified atmosphere at 37 °C, 5% CO_2_ [37]. Within 5 days of culture, the spheroids reached a diameter between 400 and 500 µm and were treated as described below.

### 2.2. Electropulsation Device and Electrical Parameters

#### 2.2.1. ECT and GET Device

Cells and spheroids were pulsed using an ELECTROCELL S20 high-voltage generator (Leroy Biotech, Toulouse, France) delivering unipolar square wave electrical pulses with a maximum voltage of 2 kV. The maximum current delivered by this generator is 8 A. The associated parallel plate electrodes were made of stainless steel, with a 4 mm inter-electrode distance. Eight pulses of 100 µs were applied at a 1 Hz frequency interval at field values ranging from 1.4 to 2 kV/cm for the ECT treatment. Ten pulses of 5 ms at 1 Hz frequency were applied at field values between 0.5 and 0.9 kV/cm for the GET protocol. The protocols are represented in Figure 1A,B.

#### 2.2.2. HV-LV Device

2D and 3D cell models were electropulsed with ELECTROCELL B15 (Leroy Biotech, Toulouse, France), with a maximum voltage of 1.5 kV for high-voltage pulses and 0.2 kV for low-voltage pulses, with a maximum current of 24 A. The associated electrodes were the same as above. HV parameters remained constant, 100 µs at 1 kV/cm, and LV parameters were set between 0.1 to 0.5 kV/cm, with pulse durations ranging from 100 to 250 ms in order to keep the product of the intensity and duration constant. One train of pulses was applied. The HV-LV protocol is schematized in Figure 1C.

### 2.3. Electropulsation

#### 2.3.1. Cells in Suspension

Cells were grown in T-25 flasks, and when they reached a confluency of about 75%, they were trypsinized, resuspended in culture medium and centrifuged 5 min at 0.3× *g*. A pool of 200,000 cells was used per assay and suspended in a low conductivity iso-osmotic pulsation buffer [24] (8.1 mM dipotassium phosphate, 1.9 mM monopotassium phosphate, 1 mM magnesium chloride in water and 250 mM sucrose; pH: 7.4, osmolarity: 270 osmol/L, conductance: 1.7 S/m), to which 100 µM of propidium iodide (PI) (Sigma Aldrich, USA) or 40 µg/mL of CpG plasmid free-encoding the tdTomato fluorescent protein (pCMV-CpGfree-tdTomato, Invivogen, Toulouse, France) was added, before pulsation using the parameters described in Section 2.2.1 and Section 2.2.2. Cells pulsed with PI were immediately analyzed by flow cytometry. Cells pulsed with plasmid were grown for 24 h in 12-well plates prior to the evaluation of the cell viability and the determination of the cell transfection rate.

#### 2.3.2. Spheroids

After being rinsed in a low-conductivity pulsation buffer, spheroids were electropulsed in 100 µL of pulsation buffer alone or containing propidium iodide at 200 µM or 40 µg/mL of pCMV-CpGfree-tdTomato. Pulses were delivered according to the protocols described in Section 2.2.1 and Section 2.2.2, and subsequently, the spheroids were observed by fluorescence or biphoton microscopy.

### 2.4. Spheroids Clearing

Spheroids pulsed with PI were directly fixed overnight with 4% PFA at 4 °C. Spheroids treated with Tomato plasmid were fixed 24 h after treatment. Once fixed, spheroids were embedded in low-melting-point agarose (QA-Agarose TM, Qbiogene, Carlsbad, CA, USA) before the following steps. Spheroids were delipidated by making three different baths: 30% tert-butanol + 3% quadfol for 2 h, 50% tert-butanol + 3% quadrol for 4 h and 70% tert-butanol + 3% quadrol for 4 h. Subsequently, a dehydration step was carried out by incubation in 70% tert-butanol-30% PEG for 24 h. Finally, to clarify spheroids, an incubation in a 75% benzyl-benzoate-25% PEG solution for 12 h was performed. Dehydration, delipidation and clarification steps were performed at 37 °C under mild agitation before the microscopy observation step [38]. The products were purchased from Sigma Aldrich, USA.

### 2.5. Results Analysis

#### 2.5.1. Flow Cytometry

Cells in suspension were diluted in 300 µL of PBS and placed on ice prior to flow cytometry analyses. The PI or Tomato fluorescent cells were analyzed by flow cytometry (BD LSRFortessa™ X-20 Cell Analyzer, Becton Dickinson FAC Scan; Becton-Dickinson Biosciences, Franklin Lakes, NJ, USA) to determine the percentage of permeabilized or transfected cells, respectively. The associated fluorescence intensity was also quantified. The excitation wavelength of the laser beam was 488 nm, and the emission wavelength was 617 nm. Data were analyzed with the FlowJo software. A region of interest was defined according to the size and granulosity of cells. Histograms representing the number of cells counted according to their fluorescence intensity were obtained [35].

#### 2.5.2. Cell Viability

A crystal violet colorimetry method was used [24]. Briefly, cells were rinsed with PBS (Ca^2+^ Mg^2+^). Each well containing cells was covered with 400 µL of 0.1% (*m*/*v*) crystal violet diluted in pulsation buffer and incubated for 15 min under mild agitation. Subsequently, the cells were rinsed three times with PBS (Ca^2+^ Mg^2+^). Finally, the cells were incubated with 400 µL of 10% acetic acid diluted in water to obtain cell lysates. Fifty µL of cell lysate of each pulsation condition were diluted with 950 µL of water, and the optical density (OD) at 595 nm was measured by spectrophotometry (Hitachi’s U-5100 UV-Visible Spectrophotometer, Santa Clara, CA, USA). OD is proportional to the number of cells. Cell viability, expressed in percentage (%V), is given by the following formula: % V = (OD)/(OD _(0 Volt)_) × 100.

#### 2.5.3. Fluorescence Microscopy

Images of the spheroids were acquired with the IncuCyte^®^ Live Cell Analysis System Microscope (Essenbioscence, Ann Arbor, MI, USA), with ×10 objective magnification. Spheroids were observed directly after electropermeabilization for PI treatment and 24 h after gene-electrotransfer with Tomato plasmid. For the spheroid growth and viability, GFP spheroids were observed for the following 5 days, and the quantification of the GFP area corresponding to the viable cells was evaluated with Image J software. We determined the viable spheroid area as the function of the area occupied by GFP-positive cells (viable cells). Concretely, on Image J (version 1.49q), we determined a threshold with a spheroid control (not treated), and we applied the same threshold to treated spheroids.

#### 2.5.4. Biphotonic Microscopy

Cleared spheroids were maintained in a 1 mm-thick imaging chamber (CoverWell, ThermoFischer Scientific, Waltham, MA, USA) filled with ethylcinnamate between the slide and the cover-slide. The observation was made under a 7MP biphoton microscope (Laser Scanning Microscopy, Zeiss, Oberkochen, Germany) equipped with a ×20 immersion objective (NA 0.95) and coupled to a Ti-Sapphire femtosecond laser, Chameleon Ultra 2 (Coherent Inc., Santa Clara, CA, USA), tuned to 800 nm and 920 nm for the PI and GFP/Tomato signals, respectively. The GFP signal was detected through a 500–550 nm band pass filter, and the PI and Tomato signals through a 565–610 nm band pass filter with non-descanned detectors. The 3D stacks of spheroids (607 × 607 µm, variable z) were acquired at a resolution of 0.59 µm and with a z-sacking of 4 µm. The resulting 3D images were stored and analyzed off-line using Imaris software 9.3 (Bitplane AG, Schlieren, Switzerland) to show both transversal and side views of the spheroids.

### 2.6. Statistics Analysis

Three independent experiments were made on one million cells’ suspension, and two biological replicates were made on 4 multicellular spheroids. Statistical analysis was performed using the GraphPad Prism 5.04 program (GraphPad Software, Inc., La Jolla, CA, USA), and data were expressed as means +/− SEM. For comparisons, the 1-way ANOVA with Bonferroni’s post-test was used, and the overall statistical significance was set at *p* < 0.05.

## 3. Results

### 3.1. Effects of ECT and GET Parameters on Cell Permeabilization, Cell Viability and Plasmid Transfection in Cells Suspensions

To quantify the electropermeabilization of HCT-116 cells in suspension, the entrance of propidium iodide was assessed by flow cytometry. Two parameters were evaluated: the percentage of fluorescent cells (i.e., propidium-iodide-positive cells), which allowed determining the permeability rate, and the fluorescence intensity, reflecting the amount of propidium iodide internalized in cells (i.e., permeabilization efficiency). Two sets of electrical conditions were tested: ECT parameters (8 × 100 µs, 1 Hz), being the standard parameters of electrochemotherapy, and GET parameters (10 × 5 ms, 1 Hz), being the optimized parameters for gene electrotransfer, as they act on both cell permeabilization and electrophoretic drag. Permeability was determined upon varying the field strength. The results (Figure 2A) showed that both protocols were highly efficient in terms of cell permeabilization. An average of 97% of permeabilized cells was obtained after ECT electropulsation. Treatment of cells with GET parameters led to similar permeability rates.

Regarding the fluorescence intensity of the PI uptake upon the ECT protocol, there was no effect when increasing the electric field intensity (Figure 2A). Indeed, the intensity varied between 70,000 a.u and 93,000 a.u, but there was no significant difference, even if the value for the ECT condition at 1.6 kV/cm was slightly higher. For the GET protocol, there was an increase of the fluorescence intensity until 0.7 kV/cm, reaching a plateau value when the electric field intensity increases. The condition at 0.5 kV/cm led to a two-times lower penetration of PI (53,078 a.u) than with 0.7 kV/cm (116,929 a.u). Comparing the two protocols, only the fluorescence intensity of the condition at 0.5 kV/cm (GET) was significantly lower than the 1.6 kV/cm ECT.

In addition to the efficient cell permeabilization, cell viability is another key parameter for gene electrotransfer and expression. Cell mortality has to be monitored and should be as low as possible if the ultimate goal is to transfer genes. Viability rates were determined for the ECT and GET protocols (Figure 2B) using crystal violet staining at 24 h. Cells pulsed with the ECT conditions between 1.4 and 1.8 kV/cm had a viability rate higher than 60%. A more pronounced cellular mortality was observed above 1.9 kV/cm (40% of viability at 1.9 kV/cm and 30% at 2 kV/cm). Following the GET parameters, with longer pulse durations, cell viability also decreased as the field intensity increased. At 0.5 and 0.6 kV/cm, 50% of cells were still viable, but the viability percentage was thus lower than under the ECT conditions. Field values from 0.7 kV/cm to 0.9 kV/cm led to the death of a large fraction of cells (viability rate of about 20%).

The same electric parameter conditions were used for plasmid DNA transfer. The transfection efficacy was characterized by measuring the fluorescence intensities by FACS analysis of the entire cell population (Figure 2C). For the ECT parameters, the higher the field strength, the higher the rate of cell transfection. Under the highest field values (1.9 and 2 kV/cm), respectively, 20% and 25% of transfected cells were obtained. However, the associated fluorescence intensity remained low, which reflected that a small amount of DNA was electrotransfered into the cells. Under the GET conditions, the rate of cell transfection also increased with the field strength from 15% to 57% by increasing the field intensity from 0.5 kV/cm to 0.7 kV/cm. Above 0.7 kV/cm, a plateau was obtained, with about 60% transfection. The fluorescence intensity also varied with the rates of transfected cells, with the highest values at 0.7 and 0.8 kV/cm (around 100,000 a.u.), which was 20 times higher than under the ECT conditions. The amount of Tomato expressed in cells at 0.9 kV/cm decreased to 65,000 a.u (Figure 2C) due to cell mortality caused by the 5 ms pulse duration at this field value (Figure 2B and images in Figure 3). Figure 3 showed a global overview of transfected cells, where only the highest florescent cells can be visualized on the micrographs.

The viability rates described in the preceding paragraph were in agreement with the images of treated cells and observed 24 h later under a fluorescence microscope (Figure 3). Indeed, the cellular layer was much less dense after ECT protocols at 1.9 and 2 kV/cm and GET protocols at 0.7 to 0.9 kV/cm, and cells exhibited a more spherical shape, indicating the cells’ gradual detachment. Figure 3 also illustrates the transfection data obtained by FACS, where it was visible that the ECT parameters allowed transfecting some cells with a lower fluorescence intensity. Visually, it was clear that GET appeared more efficient, showing a higher number of cells expressing Tomato. Finally, based on Figure 2C and the microscopy images shown in Figure 3, the GET condition at 0.6 kV/cm appeared to be the optimal parameters for a transfection rate approaching 40% (37.40% ± 3.70 SEM), with a high intensity and lower cell mortality.

### 3.2. Effects of HV-LV Parameters on Cell Permeabilization, Cell Viability and Plasmid Transfection in Cells Suspensions

The same experiments were done with “High Voltage–Low Voltage” (HV-LV) electrical parameters, where the HV remained constant and the LV varied from a lower (0.1 kV/cm) to a higher (0.5 kV/cm) intensity, with long pulse durations (500 ms) to shorter (100 ms) pulse durations. Separate HV and LV conditions were also tested (Table 1). The interest in studying HV-LV parameters stems from their potential to be more effective for gene transfection. Precisely, the HV pulses allow the permeabilization of cell membranes, and LV pulses allow a translational electrophoretic movement of plasmids through the pores induced by the previously applied HV pulse, resulting in the intracellular accumulation of plasmid DNA. The occurring electrophoretic drag would indeed be of interest in 2D-grown cells, but potentially also in 3D-grown cells, where plasmid delivery remains a major challenge. In order to compare HV-LV pulses, a constant factor of electric field intensity multiplied by pulse time was used, where E.t = 50, in order to have the same electrophoretic force and, therefore, a comparable displacement of the plasmid DNA towards the cells during pulses’ application. The uptake of PI, as previously used to assess cell permeability, clearly depended on the field intensity. Taken independently, HV alone resulted in the permeabilization of 50% of cells (Figure 4A) and induced low cell mortality, which was close to 15% (Figure 4B). At 0.1 and 0.2 kV/cm, the LV conditions alone did not lead to any cell permeabilization. Above 0.3 kV/cm, cell permeabilization was obtained, and it increased with the field strength (Figure 4A). However, the cell viability was highly affected, with only 17% of viability at 0.5 kV/cm. The HV-LV conditions allowed obtaining permeabilization even at low field intensities of LV pulses. Increasing field intensities to 0.4 and 0.5 kV/cm for LV led to permeability rates above 90% and fluorescence intensities close to 100,000 a.u., values similar to those obtained under the ECT and GET conditions. However, as seen in Figure 4B and Figure 5, the cell viability was very low with these 2 conditions, and very few cells were adhering to the support due to the 80% mortality rate. Figure 4B and Figure 5 showed that the 2 HV-LV conditions, which did not cause significant cell mortality, were 500 ms at 0.1 kV/cm (condition 4) and 250 ms at 0.2 kV/cm (condition 5). Overall, 75% percent of cells remained viable under these 2 conditions, which was dramatically higher than the 50% of cell viability obtained for the GET condition. Moreover, these 2 conditions led to a similar permeabilization rate around 80% (Figure 4A).

The protocol involving HV-LV pulses (Figure 4C) revealed that a too-weak field value in the LV regimen (condition 4 with 0.1 kV/cm) did not allow transfecting a significant percentage of cells (10%). The result was the same for LV at field values of 0.4 and 0.5 kV/cm, which caused significant cell death (Figure 4B and Figure 5): 8% (condition 7) and 6% (condition 8). The fluorescence intensities were directly proportional to the transfection rates obtained. Finally, the condition in HV-LV allowing the best viability/transfection ratio appeared to be condition 5, with an LV of 0.2 kV/cm and a pulse duration of 250 ms, which resulted in 45% with a 80,000 a.u fluorescence intensity. The microscopy images showed a large number of adherent and viable red-labeled cells (Figure 5). This condition still appeared to be significantly less efficient for transfection (*p* < 0.05) than the one at 0.6 kV/cm under the GET parameters, but still more effective than the ECT ones. Based on these results, the following step was to compare different protocols in 3D multicellular spheroids.

### 3.3. Effects of ECT, GET, HV-LV Parameters on Permeabilization and Growth of Spheroids Models

In order to assess the efficacy of different protocols in a tissue-like structure, we subsequently compared the yield of the above-mentioned pulsed electric field parameters on a 3D model, the multicellular spheroid, which resembles a small avascular tumor and is characterized by tightly adjacent cells. The spheroids were pulsed under ECT parameters at 1.6 kV/cm 100 µs, GET at 0.6 kV/cm 5 ms and HV-LV condition 5, which provided the best transfection/viability results obtained in 2D cells (Table 1). Namely, the LV pulse duration of 250 ms combined with HV allows a transfection rate similar to the GET parameter, without provoking significant cell death. Two identical sets of experiments were performed. Following the first set, the spheroids were fixed and observed with biphoton microscopy, and the second set of experiments was applied to spheroids, which were subsequently processed with the tissue-clearing technique prior to microscopy. Tissue clearing was used to allow volumetric imaging of 3D specimens, as this procedure amplifies the penetration of light within the microtissues and allows both a better penetration of excitation light entering the spheroid and the fluorescence emission to return to the detector [38,39]. This technique was used herein to ascertain that the potential lack of signal visualization under the live microscopy parameters was actually due to a lack of transfection and not due to a lack of visualization in an opaque cellular environment. The results shown in Figure 6 showed that in the set of experiments that were not cleared, only the outer layers of the spheroids (Figure 6A–C) were visible. For the three conditions tested, only the external rim of propidium iodide-labeled cells could be observed under the microscope, and no labeled cells were visible in deeper layers. On the contrary, spheroids’ clearing (Figure 6D–F) revealed that spheroids were well permeabilized from the surface to the core. These tests allowed us to validate the method of tissue clearing to deepen the visualization of internal cellular layers in our 3D model. In addition, the clearing allowed us to ascertain that the three electrical protocols lead to efficient permeabilization of all the cells constituting the spheroids. The fluorescence intensity of all cells was high, indicating that the diffusion of PI in spheroids was homogeneous.

The viability of multicellular spheroids was also assessed by evaluating their growth over 5 days (Figure 7). Spheroids were pulsed using ECT pulse parameters at 1.6, 1.8, 2 kV/cm and 100 µs, GET pulse parameters at 0.6, 0.7, 0.8 kV/cm 5 ms and HV-LV conditions 4 and 5 (Table 1). Non-treated spheroids were referred to as control. Overall, the growth of all pulsed spheroids was significantly altered compared to the controls. All spheroids treated by electrical pulses slowed down their proliferation. This observation was even more pronounced for GET-treated spheroids, whose growth curves did not evolve significantly over time. In the case of spheroids treated with GET at 0.8 kV/cm, their size decreased 24 h after treatment, and their proliferation restarted only at day 3. After 5 days, their average area reached 150,000 a.u., against 300,000 a.u. for the controls, so they were 2 times smaller. The spheroids pulsed by GET pulse parameters at 0.8 kV/cm were the most affected by the electric pulses. In the case of spheroids treated by ECT pulses, the effect of electric pulses were milder, their sizes being slightly bigger after 5 days (180,000 a.u.) compared to GET-treated spheroids (130,000 a.u.), but still significantly smaller than control spheroids. Although the spheroids treated with ECT at 2 kV/cm grew more slowly than those at 1.6 and 1.8 kV/cm, the 3 ECT protocols showed similar area values after 5 days of follow-up (180,000 a.u.). Concerning the spheroids treated with the 2 HV-LV protocols, their areas were around 150,000 a.u. at 48 h, which was comparable to that obtained directly after pulsation (day 0). No significant difference was observed for the HV-LV protocol with a LV of 500 ms at 0.1 kV/cm compared to that with a LV of 250 ms at 0.2 kV/cm. Their growths were not significantly different from the spheroid growths pulsed by ECT and GET (0.6 and 0.7 kV/cm) during 48 h. Beyond 48 h, the spheroids treated with HV-LV proliferated more rapidly and reached 200,000 a.u. after 5 days, being significantly higher than those treated by GET.

### 3.4. Effects of ECT, GET, HV-LV Parameters on Plasmid Transfection in Spheroid Models

Plasmid DNA transfection assays were also conducted on HCT-116 GFP spheroid models using the electrical parameters, which were the most efficient in suspended cells: ECT at 1.6, 1.8, 2 kV/cm; GET at 0.6, 0.7, 0.8 kV/cm and HV-LV conditions 4 and 5, as described in Table 1. Initially, spheroids were observed by bright field/fluorescence wide-field microscopy (Figure 8). The first observation was that beyond 0.6 kV/cm, spheroids pulsed with GET parameters were negatively affected by electropulsation; cells of the outer layers died and detached from the spheroid. This was also the case for ECT-pulsed spheroids. HV-LV-treated spheroids appeared smaller than control spheroids. Spheroids were observed under microscope 24 h after treatment, and the images are shown in Figure 8. Concerning plasmid transfection, the images in Figure 8 revealed that ECT pulse protocols of 100 µs at 1.6, 1.8 and 2 kV/cm allowed transfecting a few cells, especially on the outer layers of the spheroid. Overall, the number of labeled cells did not increase drastically with GET parameters compared to ECT. In addition, the transfection was not more effective under HV-LV conditions. Only 1 transfected zone was visible for the HV-LV protocol with a LV of 0.1 kV/cm at 500 ms, and with a LV of 0.2 kV/cm at 250 ms of low intensity.

Pulsed electric field parameters were chosen according to the results obtained from spheroids’ viability follow-up (Figure 7). As in the videomicroscopy images (Figure 8), we mainly focused on spheroids’ surface, and we performed tissue clarification to visualize the spheroids throughout their entire volume. Therefore, we conducted tissue clearing on the same spheroids in order to observe a possible expression of plasmids inside the inner layers of spheroids. GFP-positive cells indicate viable cells. Spheroids where clarified after gene electrotransfer and are shown in Figure 9 after ECT pulse parameters at 1.6 kV/cm, GET at 0.6 kV/cm and HV-LV with LV at 0.2 kV/cm of 250 ms. These three pulse parameters conditions were selected as they displayed transfected cells and as spheroids’ viability was affected to the smallest extent. Upon clarification, we noted that ECT and HV-LV appeared smaller than the control, probably because the cells of the outer layers detached upon spheroids’ manipulation (as the spheroids had to be transferred by pipetting to numerous solvent baths required for clarification). Spheroids treated under the GET parameters shown in Figure 9 had a size comparable to the control, but the fluorescence intensity of GFP protein was much lower compared to other conditions. The clarification of spheroids (Figure 9) showed that no cells in the inner layers of spheroids expressed the Tomato plasmid for the ECT and GET protocols. In the case of spheroids treated with ECT, the expression of plasmid was also not visible on the outer layers. It was also visible that only two cells in the outer layer of the GET-treated spheroid were transfected. Intriguingly, other transfected areas could not be visualized by microscopy after clearing. In the spheroid treated under the HV-LV condition, one labeled cell is pointed out on the outer layer, and two cells can be visualized in the inner layers on the transversal section (red arrows, Figure 9).

## 4. Discussion

Based on the results obtained on 2D cells, it was possible to compare the different electrical parameters tested for plasmid DNA transfer on the HCT-116 cell line. The ECT pulse parameters did not allow large amounts of DNA to be transfected into cells, in contrast to the GET assays, which reveal once again the importance of the pulse duration. DNA is a molecule much larger than the cytotoxic drugs used in electrochemotherapy; thus, pulses of the range of hundreds of microseconds are not sufficient to cause a significant electrophoretic drag necessary to bring DNA close to cell membranes (Figure 2C). Indeed, the DNA displacement (L) due to electrophoretic forces is given by the equation L = µ.E.t, where E is the field intensity, t is the pulse duration and µ is the electrophoretic constant of the given charge of the DNA molecule. According to the results presented, the 100 µs pulse duration in the ECT pulse parameters led to an average of 20% of transfected cells, while 50% of transfected cells were obtained with the GET pulse parameters with longer (5 ms) pulse durations. Repeated pulses of a few milliseconds were thus required, which was in agreement with previous studies [9]. Indeed, the E.t product for ECT at 1.6 kV/cm was 1.28 (E.t), while it was 30 for GET at 0.6 kV/cm, leading to a more efficient electrophoretic drag for the GET protocol. Consequently, electrophoretic drag appeared as a critical factor for efficient transfection [25,26]. A total of 10 pulses of 5 ms at 1 Hz at 0.6 kV/cm (GET at 0.6 kV/cm) appeared as the condition yielding a high percentage of transfected cells without causing considerable cell mortality (Figure 2B,C).

In addition, we wanted to compare this GET condition, defined as an “efficient control”, with the HV-LV pulse parameters. We have chosen conditions where only the LV varies, so that the field value (E) that multiplies the LV pulse duration (t) was always equal to 50, in order to have an equivalent electrophoretic drag/force. This electrophoretic drag was set to be slightly higher than that of the GET protocol (E.t value is 30) in order to increase the potential interaction with the cells in spheroids. A total of 100 µs at 1 kV/cm (HV)—500 ms at 0.1 kV/cm (LV) and 100 µs at 1 kV/cm (HV)—250 ms at 0.2 kV/cm (LV) led to an efficient permeabilization and were the conditions that did not cause important cell mortality (less than 15%), but only the condition with a 250 ms at 0.2 kV/cm as LV significantly transfected cells (45%) (Figure 4A–C). In addition, in accordance with literature observations [27,29], which asserted that a short high-voltage pulse allowed cell permeabilization, and a longer low-voltage pulse allowed DNA transfer by electrophoresis, our results showed the same on HCT-116 cells. Herein, cell permeability was observed after all the combinations of HV-LV. In our study, the HV alone led to 50% of permeabilization (Figure 4A), while Satkauskas and colleagues observed a more important permeability rate with a field value of 0.8 kV/cm [30]. The HV-LV condition, which was the most effective during our tests (100 µs at 1 kV/cm and 250 ms at 0.2 kV/cm), allowed significant permeability, while independent LV had no effect. Independent LV inefficiency in terms of cell permeabilization was in accordance with previous studies [25,29]. Even if it provoked smaller cell mortality (Figure 4B), the combined HV-LV protocol (condition 5) appeared to be less effective than GET in terms of DNA transfection on suspended cells, as evidenced by the percentage of transfected cells and the associated fluorescence intensity (Figure 4C).

Major differences could be observed in 3D compared to the suspended cells tested for gene transfer. As previously shown [24], spheroids are more complex than 2D cells and, thus, were more difficult to transfect. A tiny number of cells was transfected and only on the outer layer of the spheroid, which is in accordance with previous studies [34]. Taken together, our results show that electrotransfer is poorly efficient, regardless of the pulsing protocol used (Figure 8 and Figure 9). These results were observed for all electrical protocols, even GET, which was actually efficient on suspended cells, with transfection percentages around 50% with high-intensity pulses (Figure 2C). This could be explained by the accessibility of 2D cells suspended and scattered in the pulsing buffer. As for spheroid cells, they adhere to each other by strong adherent junctions and were not easily accessible to the plasmid. Another explanation could be relative to the cellular stage in the spheroids. Due to nutrient and oxygen gradients, cells in spheroids’ core might be apoptotic or necrotic [40] and, consequently, not able to express plasmid. In addition, proliferative cells on the outer layers, which corresponded to the potentially transfectable cells, were more sensitive to the electric field and appeared to die. Indeed, our results assessed that the application of electric field (ECT, GET and HV-LV protocols) affected the growth of spheroids (Figure 7) and contribute to the detachment of cells on spheroids’ surface (images Figure 8). The method of clearing optimized on spheroids [41,42,43] and on 3D tissues in a more general way [44] allowed us to confirm that all cells constituting the spheroids underwent electropermeabilization, regardless of the protocols applied. The same tissue-clearing procedure was used in our experiments to study DNA transfection in deeper layers. The results confirmed that the plasmid encoding tdTomato (MW: 3 MDa), which was much larger than propidium iodide (MW: 668 Da), probably did not diffuse into this 3D cellular model. In cleared spheroids, we observed less tdTomato-expressing cells. A possible explanation could be that transfected cells detached during the pipetting protocols, when spheroids were transferred to different solvent baths. Conversely, in the case of propidium iodide, the fluorescence was detectable by microscopy across the entire spheroid. Taken together, the clearing method is innovative and has never been used before to observe the penetration of molecules into spheroids after the application of the electric field. The method efficiently evidenced the permeabilization of 3D models, allowing the visualization of permeabilized cells, even in the core of the spheroids. Concerning gene transfer, future studies are necessary to assess if the observed phenomena were due to a lack of penetration of plasmid DNA or a lack of plasmid expression, or due to the quiescence of the cells located in the spheroids’ core. The distribution will be assessed in further studies, where we will use a fluorescently labeled plasmid to directly visualize its distribution and prior gene expression.

## 5. Conclusions

This study leads to three main conclusions:(i)Transfection of suspended cells is efficient after GET parameters at field values between 0.6 and 0.8 kV/cm, as well as with HV-LV parameters with a LV of 0.2 kV/cm at 250 ms to avoid excessive cell mortality.(ii)Transfection of spheroids is much less effective and occurs on proliferative external cells. Spheroids are difficult to transfect and are, as such, a relevant model for predicting in vivo outcomes.(iii)Clearing of spheroids shows that inner cells are efficiently electropermeabilized under all the tested protocols, but not transfected, potentially indicating the problem of plasmid DNA diffusion in tissues and solid tumors. This technique opens new perspectives to study and quantify the delivery of molecules in tissues.

## Figures and Tables

**Figure 1 pharmaceutics-15-01004-f001:**
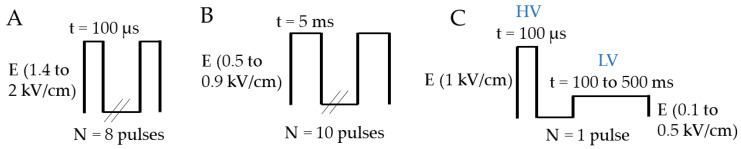
Schematic representation of electrical parameters used. (**A**) ECT pulse protocol (8 pulses of 100 µs at 1.4 to 2 kV/cm, 1 Hz), (**B**) GET pulse protocol (10 pulses of 5 ms at 0.5 to 0.6 kV/cm, 1 Hz) and (**C**) HV-LV pulse protocols with HV constant (100 µs at 1 kV/cm) and LV variant (500 to 100 ms at 0.1 to 0.5 kV/cm).

**Figure 2 pharmaceutics-15-01004-f002:**
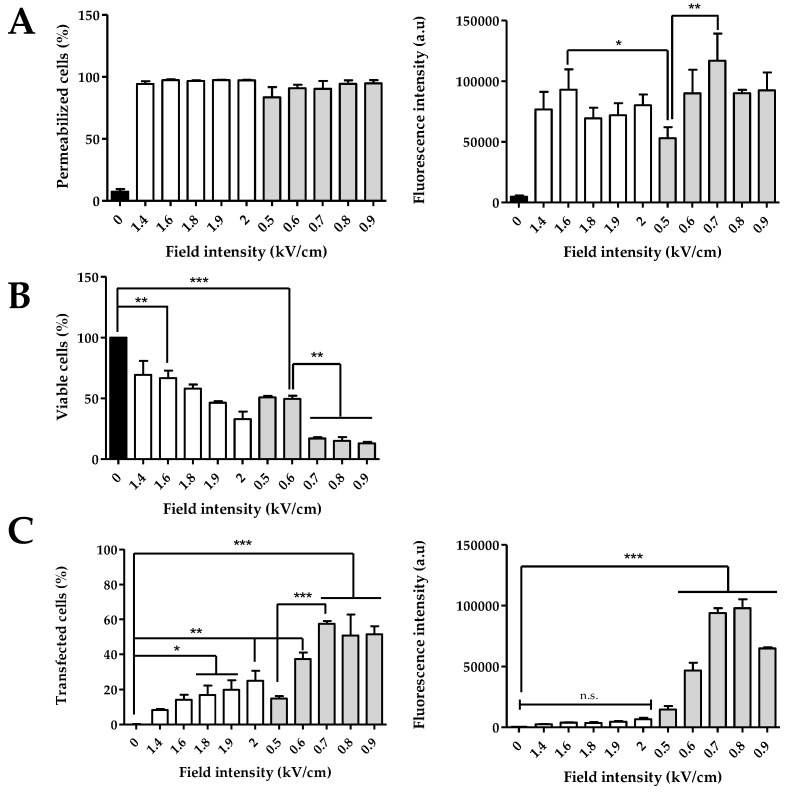
Effect of GET and ECT pulse parameters on HCT-116 cells as function of field intensity. (**A**) Percentages of permeabilization and associated fluorescence intensity, (**B**) cells viability, (**C**) percentages of transfected cells and associated fluorescence intensity. White histograms: cells treated with ECT pulse parameters (8 pulses of 100 µs at 1 Hz), Light gray histograms: cells treated with GET pulse parameters (10 pulses of 5 ms at 1 Hz). Cells were incubated with PI 50 µM for electropermeabilization (**A**) and plasmid encoding tdTomato 40 µg/mL for gene electrotransfer (**C**). N = 3 independent experiments. One-way ANOVA test was performed. *** *p* < 0.001, *** p* < 0.01, * *p* < 0.05.

**Figure 3 pharmaceutics-15-01004-f003:**
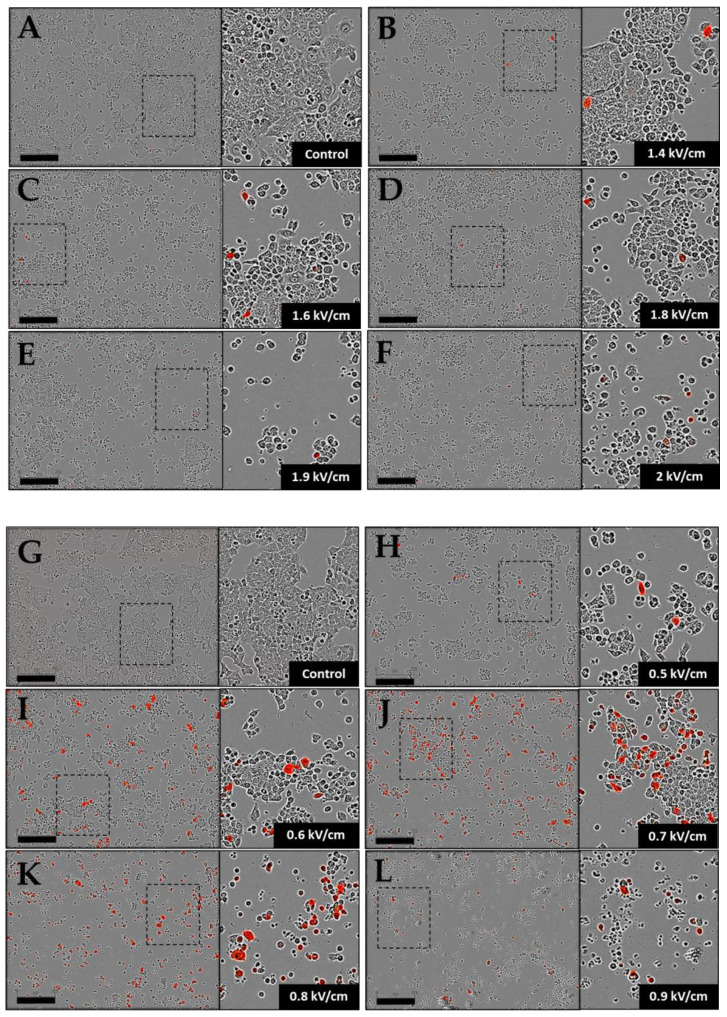
Representative bright field and fluorescent micrographs of HCT-116 cells, observed 24 h after pulsation. Cells were incubated with 40 µg/mL plasmid-encoding tdTomato, and ECT pulse parameters (8 pulses of 100 µs at 1 Hz with variables field intensity (1.4 to 2 kV/cm)) and GET pulse parameters (10 pulses of 5 ms at 1 Hz with variables field intensity (0.5 to 0.9 kV/cm)) were applied. (**A**,**G**) Control; (**B**–**F**) ECT pulse parameters and (**H**–**L**) GET pulse parameters. Cells expressing tdTomato emitted red fluorescence 24 h after pulsation. Scale bar = 300 µm. Zoom on the right corresponds to the rectangle in the entire image.

**Figure 4 pharmaceutics-15-01004-f004:**
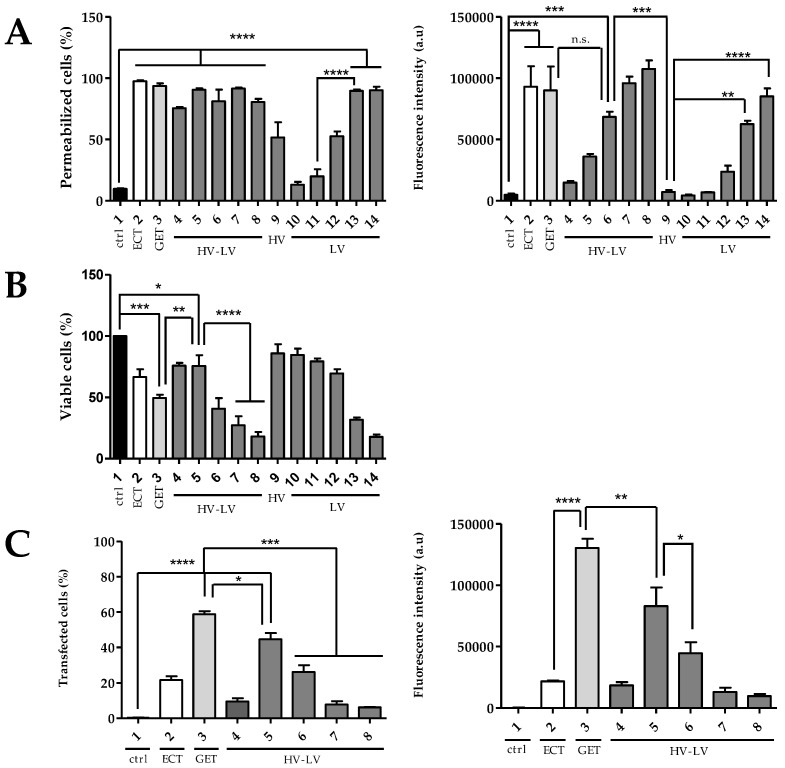
Effect of HV-LV pulse parameters on HCT-116 cells as function of field intensity. (**A**) Percentages of permeabilization and associated fluorescence intensity, (**B**) cells viability, (**C**) percentages of transfected cells and associated fluorescence intensity. White histograms: cells treated under ECT pulse protocol (8 pulses of 100 µs at 1 Hz), Light gray histograms: cells treated under GET pulse protocol (10 pulses of 5 ms at 1 Hz) and dark gray histograms: cells treated under HV-LV (HV constant (100 µs at 1 kV/cm) and LV variant (500 to 100 ms at 0.1 to 0.5 kV/cm)). Cells were incubated with PI 50 µM for electropermeabilization (**A**) and plasmid-encoding tdTomato 40 µg/mL for gene electrotransfer (**C**). N = 3 independent experiments. One-way ANOVA test was performed. **** *p* < 0.0001, *** *p* < 0.001, *** p* < 0.01, * *p* < 0.05.

**Figure 5 pharmaceutics-15-01004-f005:**
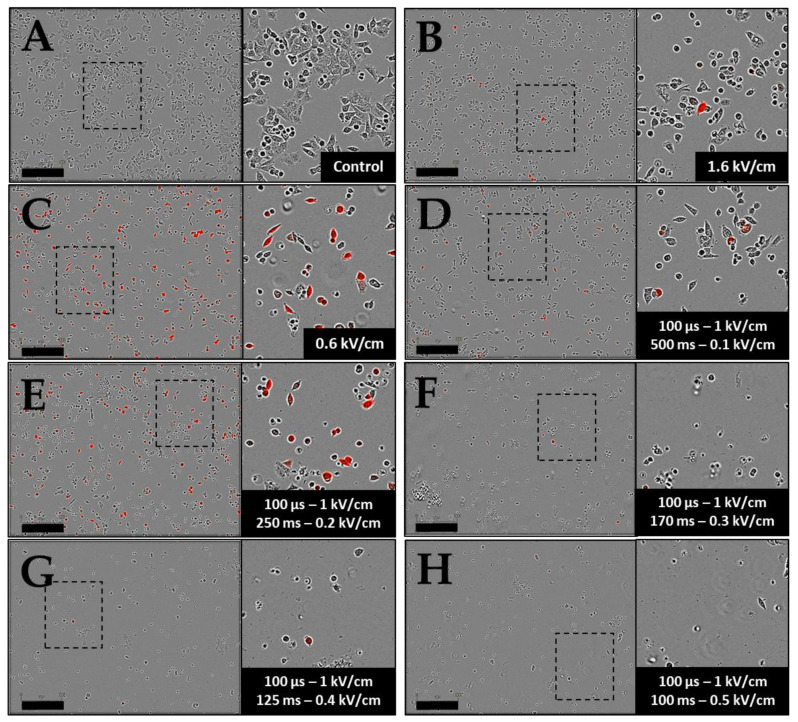
Representative HCT-116 cells electrotransfection micrographs. Cells were observed 24 h after pulsation by bright field/fluorescence wide-field microscopy. Cells were incubated with 40 µg/mL plasmid encoding tdTomato, and ECT pulse parameters (8 pulses of 100 µs at 1 Hz with variables field intensity (1.4 to 2 kV/cm)), GET pulse parameters (10 pulses of 5 ms at 1 Hz with variables field intensity (0.5 to 0.9 kV/cm)) and HV-LV parameters with HV constant (100 µs of 1 kV/cm) and variable LV (500 ms of 0.1 kV/cm; 250 ms of 0.2 kV/cm; 170 ms of 0.3 kV/cm; 125 ms of 0.4 kV/cm and 100 ms of 0.5 kV/cm) were applied. (**A**) Control, (**B**) ECT pulse parameters, (**C**) GET pulse parameters, (**D**–**H**) HV-LV pulse parameters. Cells expressing tdTomato emitted red fluorescence 24 h after pulsation. Scale bar = 300 µm. Zoom on the right corresponds to the rectangle in the entire image.

**Figure 6 pharmaceutics-15-01004-f006:**
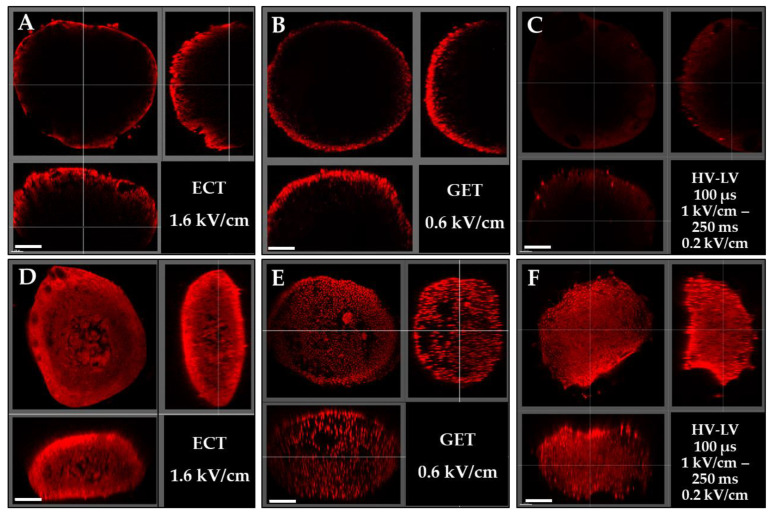
Representative 3D images of HCT-116 GFP spheroids electropermeabilization. Spheroids were incubated with 200 mM PI and treated with ECT pulse parameters (8 pulses of 100 µs, 1.6 kV/cm at 1 Hz) (**A**,**D**), GET pulse parameters (10 pulses of 5 ms, 0.6 kV/cm at 1 Hz) (**B**,**E**) and HV-LV pulse parameters (100 µs at 1 kV/cm—250 ms at 0.2 kV/cm) (**C**,**F**). One hour after treatment, spheroid were fixed and observed by biphotonic microscopy (**A**–**C**) and compared with spheroids that underwent tissue clearing (**D**–**F**). Top left: front view, top right: transversal view, up left: sagittal view on an optical plane inside the spheroid. Scale bar = 200 µm. N = 2 independent experiments, 4 spheroids per experiment.

**Figure 7 pharmaceutics-15-01004-f007:**
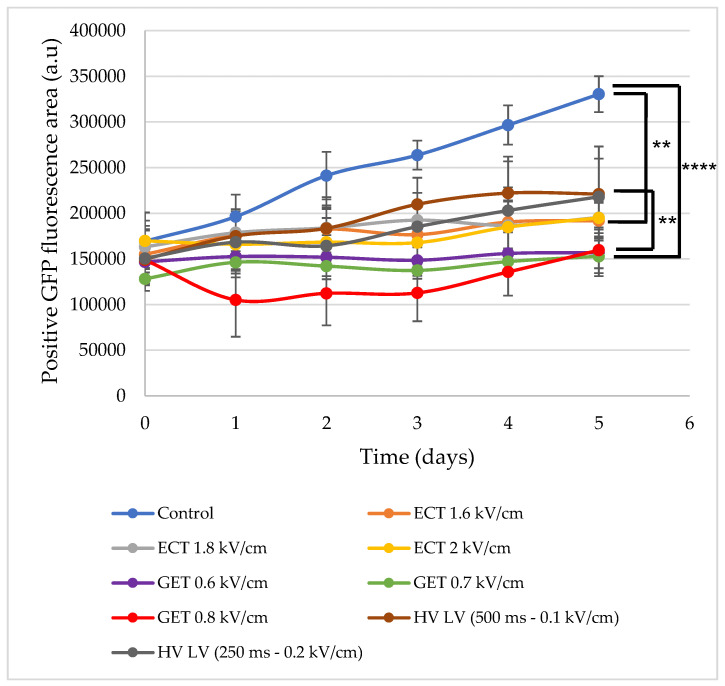
Multicellular spheroids growth as a function of time. Spheroids were pulsed with ECT pulse parameters (green, dark blue, purple curves) (8 pulses of 100 µs at 1Hz with 1.6; 1.8 and 2 kV/cm), GET pulse parameters (gray, light blue, pink curves) (10 pulses of 5 ms at 1 Hz with 0.6; 0.7 and 0.8 kV/cm) and HV-LV (red and orange curves) with HV constant (100 µs of 1 kV/cm) and LV variable (500 ms of 0.1 kV/cm and 250 ms of 0.2 kV/cm). Positive GFP fluorescence area (a.u.) was quantified every day during 5 days (expressed as mean +/− SD). N = 6 spheroids per condition. One-way ANOVA test was performed. **** *p* < 0.0001, *** p* < 0.01.

**Figure 8 pharmaceutics-15-01004-f008:**
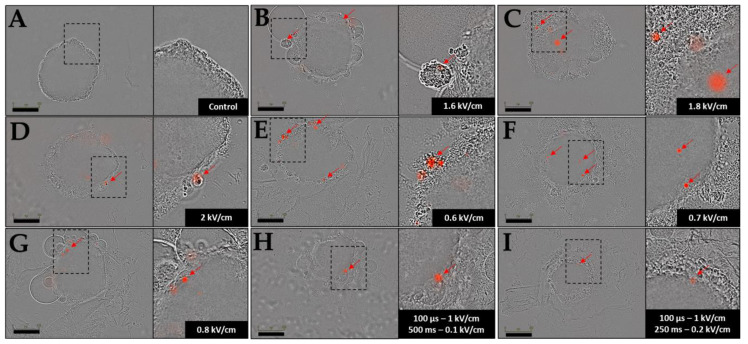
Representative HCT-116 spheroids images following the transfection attempt. Spheroids were incubated with 40 µg/mL plasmid encoding tdTomato and observed 24 h after pulsation by bright field/fluorescence wide-field microscopy. Transfected areas are indicated with red arrows. (**A**): control; (**B**–**D**): ECT pulse parameters were used (8 pulses of 100 µs at 1 Hz with 1.6; 1.8; 2 kV/cm); (**E**–**G**): GET parameters were used (10 pulses of 5 ms at 1 Hz with 0.6; 0.7 and 0.8 kV/cm) and (**H**,**I**): HV-LV with HV constant (100 µs of 1 kV/cm) and LV variable (500 ms of 0.1 kV/cm and 250 ms of 0.2 kV/cm). Scale bar = 300 µm. N = 3 independent experiments, 4 spheroids per experiment. Zoom on the right corresponds to the magnified view of the rectangle in left-adjacent panel.

**Figure 9 pharmaceutics-15-01004-f009:**
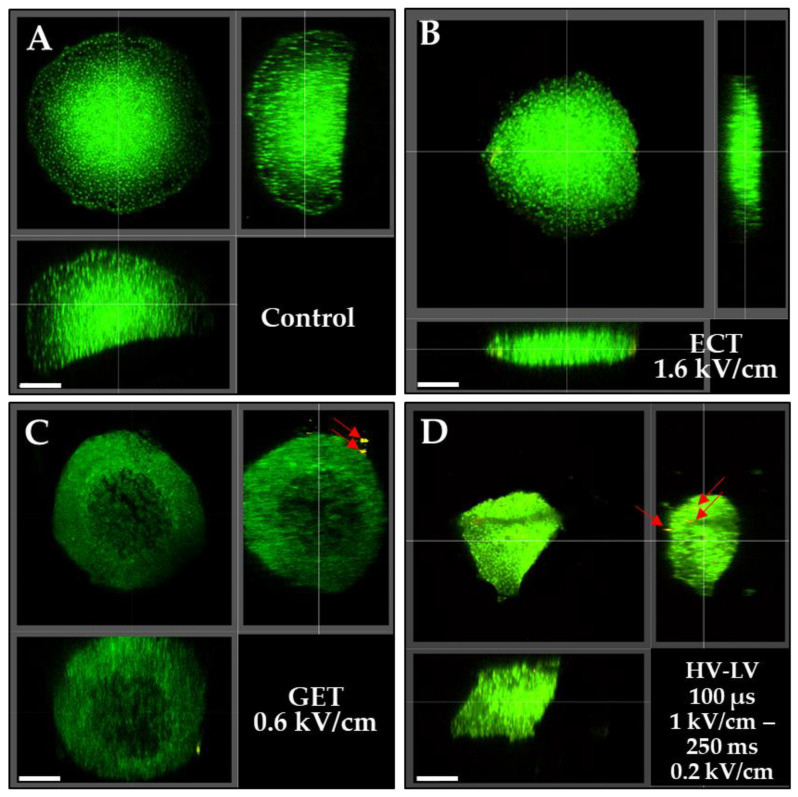
Representative 3D fluorescence micrographs of clarified HCT-116 GFP spheroids after gene electrotransfer. Spheroids were incubated with 40 µg/mL plasmid encoding tdTomato and no treated (**A**), or treated with ECT pulse parameters (8 impulsions of 100 µs, 1.6 kV/cm at 1 Hz) (**B**), GET pulse parameters (10 pulses of 5 ms, 0.6 kV/cm at 1 Hz) (**C**) and HV-LV pulse parameters (100 µs at 1 kV/cm—250 ms at 0.2 kV/cm) (**D**). Fixed spheroids were observed 24 h after pulsation. Transfected area is indicated with a red arrow. Top left: front view, top right: transversal view, up left: sagittal view of an optical plane inside the spheroid. Scale bar = 200 µm. N = 2 independent experiments, 4 spheroids per experiment.

**Table 1 pharmaceutics-15-01004-t001:** HV-LV pulse parameters used in Figure 4.

Essay	Number of Pulses	HV		LV	
		Field Intensity (kV/cm)	Pulse Duration (µs)	Field Intensity (kV/cm)	Pulse Duration (ms)
1	0	0	0	0	0
2	8	1.6	100	0	0
3	10	0	0	0.6	5
4	1	1	100	0.1	500
5	1	1	100	0.2	250
6	1	1	100	0.3	170
7	1	1	100	0.4	125
8	1	1	100	0.5	100
9	1	1	100	0	0
10	1	0	0	0.1	500
11	1	0	0	0.2	250
12	1	0	0	0.3	170
13	1	0	0	0.4	125
14	1	0	0	0.5	100

## Data Availability

All the data associated with this study are present in the paper. Materials are available upon request from the corresponding author.

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
