# Peer review of "Gene Electrotransfer Efficiency in 2D and 3D Cancer Cell Models Using Different Electroporation Protocols: A Comparative Study"

_pharmaceutics, 2023, doi:10.3390/pharmaceutics15031004_

Round 1

Reviewer 1 Report

Summary: The authors present an interesting work where gene electrotransfer efficiency in 2D and 3D cancer cell models 2 using different electroporation protocols were comparated. The manuscript is well prepared. The results presented in this study could be used by other researchers while designing the experiments especially associated with gene transfer using electroporation method.

The work is applauded especially in terms of application viewpoint. The current version requires few minor changes before formal acceptance.

1. Figure 7 is not very clear in terms of readability. Please correct this.

2. Avoid using different font sized in the subfigures. for e.g., Figure 3F and 3G

3. Figure 7: Y-axix is area or intensity? 

4. Authors are requested to proof-read manuscirpt for minor errors and consistency. 

Reviewer 2 Report

The authors have comprehensively compared different pulsed electric field protocols for gene electrotherapy and electrochemotherapy and their efficiency in 2D and 3D cellular organizations. The results of the study highlight the significance of electric field intensity, the timing of cell permeabilization, and the duration of pulses in the gene delivery process. It is particularly noteworthy that the gene-electrotherapy protocol was most efficient in cell suspensions, with a transfection rate of around 50%, while none of the tested protocols allowed for gene delivery beyond the rims of multicellular spheroids. Overall, this study makes a valuable contribution to the field of gene transfer and its potential applications in cancer therapy. I would be happy to accept this paper for publication.

Reviewer 3 Report

The Authors compare different DNA permeabilization/electro transfer protocols in the HCT-116 cell line in both a 2D and 3D model. Furthermore, the Authors provide further evidence that a clarification protocol is suitable for the visualization of fluorescent signals in the inner core of HCT-119 spheroids. The authors have long experience in electro transfer research. The present work appears to be largely confirmatory of previous work on both permeabilization/transfection and spheroid clearing. The real innovations will be brought about by the distribution studies that the authors state they intend to undertake in the near future. Some defects should be corrected as follows:

Fig 2B. The crystal violet is an indirect measure of dead cells and not very precise. I think it would be better to use a FACS analysis with PI or 7-AAD and fluorescent Annexin-5. The authors state that the optimal transfection condition is 0.6 GET, based on Fig. 2B and Fig. 3. However, the percentage of transfected cells in Fig. 3 seems to be much lower than 40% of the attached cells. Furthermore, 0.5 and 0.6 are identical in Fig. 2B and quite different in Fig. 3.

Line 275. 10,000 should be 100,000.

Line 275. The authors state that 0.9 kV/cm GET results in a reduction of fluorescence to 6,500 a.u. whereas in Fig. Fig. 2C it appears to be above 50,000 a. u.. The reference to Fig. 2C is missing.

Table 1. It seems to me that the values for GET and ECT are reversed. The 0.6 value is attributed to ECT and 1.6 to GET.

Row 361. Condition 4 shows a much higher permeability than the 28% shown in the graph (Fig. 4A). It actually appears to be 75% and 5 about 90%.

Fig. 6 and 9. GET is written EGT in boxes B and E. It is necessary to describe how the area is calculated.

Fig. 7. The areas all start from the same value at time 0, which is quite unusual unless the values are reported as percentages of day 0.

Row 447-448. After 5 days in Fig. 7 the value appears to be above 150,000 a.u. on a linear scale and not 130,000 a.u.

Fig. 8. The arrows are red and not black as indicated in the legend.

Line 505-511. Difficult to understand

Fig. 8 and 9. According to Fig. 7, HV-LV treated spheroids should be larger than GET at 24 hours, why are they actually smaller?

The authors report that they did three experiments on cells in suspension but do not state how many replicates they did, neither in the Methods section nor in the legends of Figures 2 and 4. It appears that only one replicate was made for each experiment.

Punctuation and English to be revised
